# e-THERAPIST: *I suggest you to cultivate a mindset of positivity and nurture uplifting thoughts*

**Kshitij Mishra** * and **Priyanshu Priya** * and **Manisha Burja** and **Asif Ekbal**
Department of Computer Science and Engineering
Indian Institute of Technology Patna, India
{kshitij_1921cs23,priyanshu_2021cs26,burja_2211cs05,asif}@iitp.ac.in

## Abstract

The shortage of therapists for mental health patients emphasizes the importance of globally accessible dialogue systems alleviating their issues. To have effective interpersonal psychotherapy, these systems must exhibit politeness and empathy when needed. However, these factors may vary as per the user's gender, age, persona, and sentiment. Hence, in order to establish trust and provide a personalized cordial experience, it is essential that generated responses should be tailored to individual profiles and attributes. Focusing on this objective, we propose **e-THERAPIST**, a novel polite interpersonal psychotherapy dialogue system to address issues like depression, anxiety, schizophrenia, etc. We begin by curating a unique conversational dataset for psychotherapy, called PSYCON. It is annotated at two levels: (i) dialogue-level - including user's profile information (gender, age, persona) and therapist's psychotherapeutic approach; and (ii) utterance-level - encompassing user's sentiment and therapist's politeness, and interpersonal behaviour. Then, we devise a novel reward model to adapt correct polite interpersonal behaviour and use it to train **e-THERAPIST** on PSYCON employing NLPO loss. Our extensive empirical analysis validates the effectiveness of each component of the proposed **e-THERAPIST** demonstrating its potential impact in psychotherapy settings[1].

## 1 Introduction

The prevalence of psychological and mental disorders, such as depression, anxiety, stress and others, is increasing globally. Approximately 5% of the adult population worldwide is estimated to experience depression (WHO, 2023). Consequently, the demand for counseling services continues to rise, and the existing mental health workforce is struggling to meet the needs adequately. Hence, dialogue systems possessing social influence skills like psychotherapy are crucial for expanding the application of technology to a wide range of realistic situations.

Politeness has shown to be effective in driving a smooth and engaging conversation during psychotherapy (Budiarta et al., 2021). However, politeness, in itself, encompasses various aspects (Kitamura, 2000; Laplante and Ambady, 2003; Stephan et al., 2010), making it challenging to instil in a dialogue system. For instance, the degree of politeness in communication varies based on the gender and age of the person involved (Danescu-Niculescu-Mizil et al., 2013; Mahmud, 2013; Firdaus et al., 2022b); conversations with females/elders often include more polite expressions than males/youths. Politeness can also be influenced by various aspects of an individual's personality traits (Goldberg, 1993; Hirsh et al., 2010; Xafizovna and Boboqulovna, 2022).

Interpersonal behaviour advances the understanding of the interpersonal dispositions associated with psychological problems. The interpersonal circumplex (IPC) model assesses these dispositions and facilitates the comprehension of user's social cognition, motivation, and behaviour (Locke et al., 2017). It indicates that adults and youth facing psychological issues are more likely to seek negative feedback and criticism in their interactions compared to older individuals. Further, females tend to initiate more interpersonal stressors than males, and their responses to these stressors often involve rumination and negative thinking patterns. The IPC model can also help identify patterns and relationships between personality traits, thereby fostering an understanding of how individuals interact with others (Gurtman, 2009).

The incorporation of polite and interpersonal

---

*The authors are jointly first authors.

[1]The dataset and code can be accessed at https://github.com/Mishrakshitij/e-THERAPIST.git or https://www.iitp.ac.in/āi-nlp-ml/resources.html#e-THERAPIST

**Elderly Female Openness**

*T:* A very good morning madam! How are you feeling today? (helpful)

*U:* Good morning, I'm feeling too down today. Can you help me? (negative)

*T:* I'm really sorry to hear that. Kindly tell me what's been going on? (understanding)

*U:* I just feel like nothing is going right in my life. I don't have any motivation to do anything. (negative)

*T:* I see, but don't worry, together we will explore the solutions that would be better for you. Could you please tell me more about what's been going on? (empathetic)

**Elderly Male Openness**

*T:* Good morning sir, what brings you here? (helpful)

*U:* I'm not feeling good these days. I seek your help. (negative)

*T:* Sorry to learn this. I want to know what's been going on to help you? (understanding)

*U:* I don't feel like anything right in my life, no motivation, nothing. (negative)

*T:* Okay, let's explore that. Can you please explain a bit about what you are exactly experiencing these days? (helpful)

**Young Female Openness**

*T:* Hello dear! I'm ready to provide you with empathetic support. How are you doing today? (helpful)

*U:* Not so good, not so energetic. (negative)

*T:* That's really disappointing to hear, my dear friend. I presume you are facing some issues these days. I would love to help you if you could share your problem with me. (empathetic)

*U:* I have this overwhelming sense that everything is going wrong in my life. I lack the drive and enthusiasm to engage in any activities. (negative)

*T:* Hey girl, I understand your situation a bit. Let's dive deeper to find the strategies that would be helpful. If you agree, I would like to ask a few questions? (empathetic)

**Elderly Male Agreeableness**

*T:* Welcome to our session. What brings you here today? (helpful)

*U:* I've been feeling lethargic these days. Please help me. (negative)

*T:* I apologize for the odd situation you're facing. Can you please share more details about what has been happening? (helpful)

*U:* Yes, of course. I can share with you. (positive)

*T:* I appreciate your willingness to share. Would you mind elaborating on your current experiences? (understanding)

*U:* I've been struggling to find any positivity in my life. It's disheartening to lack motivation and feeling empty. (negative)

*T:* We truly understand your situation. I will do our best to help you. Let's explore the techniques to cope with your circumstances. (empathetic)

Figure 1: Example demonstrating the polite and interpersonal behaviour of the therapist (*T*) based on the user's (*U*) gender, age, persona and sentiment. The text in green depicts the user's personal profile (gender, age and persona). The text in pink, blue and grey depict the user's sentiment, the therapist's polite and interpersonal behaviour, respectively.

conduct into the dialogue agents establishes a friendly and congenial environment, thereby, providing a better personalized experience to users. The user's sentiment further facilitates the generation of contextually correct polite (Firdaus et al., 2022a) and appropriate interpersonal behavioural responses. For instance, in Figure 1, for the *"Young Female Openness"* sample (fourth utterance), the user expresses a negative sentiment, thus agent's behaviour in the form of imposing or confrontation might make the user more angry or dissatisfied. Hence, it should adapt helping, understanding or empathetic behaviour. Besides, a therapeutic dialogue system needs to utilize various psychotherapeutic approaches (Smith, 1982; Snyder, 1945; Thorne, 1948; Howard et al., 1986) during the ongoing conversation based on user's personal profile (gender, age and persona) to produce effective outcomes during psychotherapy as depicted in Figure1. Driven by these considerations, in this work, we propose **e-THERAPIST**, a novel gender, age, persona and sentiment-aware polite and interpersonal dialogue system for psychotherapy.

To develop **e-THERAPIST**, we exploit the newly designed seven rewards in a reinforcement learning (RL) setting (Casanueva et al., 2018; Mesgar et al., 2020; Lambert and von Werra, 2023). This allows **e-THERAPIST** to learn and enhance their performance based on the rewards received through interactions with the environment. To build **e-THERAPIST**, we first curate a novel conversational dataset, named PSYCON by employing well-designed *prompts* with manual interventions. Then, we annotate the user's utterances with sentiment, and agent's utterances with politeness and interpersonal behavioural information. First, PSYCON is used to fine-tune a large language model (LLM) in a supervised setting. Then, we fine-tune this trained LLM in an RL framework incorporating a novel reward function. This reward function ensures appropriate psychotherapeutic approach, politeness, and interpersonal behaviour based on gender, age, persona and sentiment of the user along with context adequacy and fluency in the generated responses. This reward is used in a Natural Language Policy Optimization (NLPO) loss (Ramamurthy et al., 2022) to optimize the model fine-tuned in a supervised setting. Finally, the performance of the proposed system is evaluated through both automatic metrics and human assessment.

In summary, the *key* contributions of our current work are summarized as follows: **(i)** Introduced **e-THERAPIST**, a novel gender, age, persona and sentiment-aware polite and interpersonal dialogue system for psychotherapy, fine-tuned in an RL environment; **(ii)** Created a novel conversational dataset for psychotherapy, PSYCON, and annotated it at two distinct levels - (a) dialogue-level with gender, age, persona, and psychotherapeutic approach (b) utterance-level information, *viz.* sentiment, politeness, and interpersonal behaviour information; **(iii)** Devised a novel reward function incorporating five attribute-specific rewards and two response-quality rewards to generate engaging, fluent, and interactive responses tailored to the user's sentiment, gender, age, and persona; **(iv)** Conducted extensive empirical evaluation to test the efficacy of **e-THERAPIST** in terms of novel metrics, *attribute-success* and *response-quality* with respect to the strong baselines.

## 2 Related Work

The issue of mental health disorders, which is a significant concern for public health (Jacob, 2012), has been the focus of previous research, including

computational studies. While depression has received the most attention, other mental illnesses like anxiety, schizophrenia, post-traumatic stress disorder, suicide risk, and self-harm have also been examined (Uban et al., 2021). A few studies have examined the posts and blogs of users on social sites to detect depression (Yates et al., 2017; Tadesse et al., 2019), suicidal thoughts (Zirikly et al., 2019; Cao et al., 2019), and other mental health issues (Xu et al., 2020) using natural language processing (NLP) techniques. Some researchers have also worked on developing "therapybots" and creating dialogue agents to provide therapeutic support (Fitzpatrick et al., 2017).

In recent times, generating empathetic responses in psychotherapeutic conversations has grown in popularity (Morris et al., 2018; Sharma et al., 2020). In order to help mental health supporters, (Sharma et al., 2021) investigated empathy rewriting as a text generation task. The authors in (Saha et al., 2022) focused on generating sentiment-driven motivational responses in mental health support dialogues. A few studies have explored the role of politeness in improving the sense of empathy and compassion during conversation (Lucas et al., 2014; Kim et al., 2018). The agent's courteous attitude conveys a sense of concern and emotional involvement like a human companion. Further, comprehending and demonstrating proper interpersonal behaviour has proved its effectiveness in psychology to study interpersonal processes, personality traits, and relationship functioning (Kiesler and Auerbach, 2003; Pincus and Gurtman, 2006; Locke et al., 2017). Studies have suggested that good quality interpersonal relationships and behaviour are important for peoples' social functioning and mental health (Cremers et al., 2021). Lately, the authors in (Firdaus et al., 2022b) have demonstrated that inculcating politeness in the agent based on the user's personal profile, such as gender and age group makes the dialogue agent capable of identifying subtle language changes while conversing with different users. Likewise, taking the users' persona into account while generating responses will further enhance the personalization quotient in the dialogue systems (Firdaus et al., 2020; Nargund et al., 2022; Ahmad et al., 2023; Zhao et al., 2023).

Inspired by the significance of politeness and interpersonal relationship in psychological support, together with the subtle change in these aspects with users' profiles (gender, age and persona) and sentiment information, we propose a polite and interpersonal dialogue system for psychotherapy that generates responses in accordance to user's age, gender, persona and sentiment. To the best of our knowledge, ours is the first attempt that exploits politeness and interpersonal relationship to generate precise responses in dialogue systems for psychotherapy. Furthermore, our research pioneers the exploration of how politeness and interpersonal relationships differ across individuals of varying gender, age groups, and personas within psychotherapeutic conversations.

## 3 Dataset

To develop **e-THERAPIST**, we create PSYCON, a novel high-quality conversational dataset for psychotherapy. We focus on conversations considering the gender, age and persona of the user with the ultimate purpose of enhancing mental health support in a personalized way and improving the overall outlook of people facing psychological issues.

### 3.1 Dataset Creation

PSYCON comprises interactions between the therapist and the user suffering from one of the seven most common psychological issues, *viz.* depression, anxiety, stress, bipolar disorder, disruptive behaviour and dissocial disorders, post-traumatic stress disorder (PTSD), and schizophrenia (WHO, 2022). To minimize the requirement of expensive and scarce human resources, we create the dataset by utilizing knowledge present in LLMs like GPT-J model (Wang and Komatsuzaki, 2021). We create the dataset by prompting this GPT-J model followed by manual intervention to ensure quality control. The dataset creation process involves two steps: (i). Therapist-user dialogue creation, and (ii). Data cleaning and quality control.

#### 3.1.1 Therapist-user Dialogue Creation

We create therapist-user dialogues utilizing the following steps:

**Attaining the Seed Utterance.** We require seed user utterances accompanied by a specific gender, age and persona followed by the seed utterance of the therapist adhering to a designated psychotherapeutic approach to begin the few-shot dialogue generation using GPT-J model. We refer to several authentic websites, such as the World Health Organization (WHO, 2022), the National Mental Health Foundation (MHF, 2023), and the National

Alliance on Mental Illness (NAMI, 2023) to understand the specific characteristics of different psychological issues. We also utilize the real user interactions from threads posted on different mental health-focused subreddits (e.g. *r/depression*) to gather the real experiences of the users facing psychological issues.

Further, to enhance user interaction and satisfaction, it is important for the therapist to respond politely and display appropriate interpersonal behaviour. Thus, we utilize the information gathered from the mentioned sources to create the seed utterances considering the variation in politeness and interpersonal behaviour across different genders, age groups and personas while complying with a particular psychotherapeutic approach. This is done with the help of six human experts having postgraduate qualifications in English Linguistics and proficiency in politeness concepts, interpersonal behaviour theory and psychotherapeutic approaches under the guidance of a leading psychotherapist from a government-run institution. The human experts were given proper instructions for designing the seed utterances: *(a)* create a sequence of seed utterances for each possible combination of gender, age, persona and psychotherapeutic approach for a particular psychological problem; *(b)* attempt to seek information about the problem and make advances towards solution following a designated psychotherapeutic approach according to the user profile; *(c)* formulate responses concerning the variation of politeness and interpersonal behaviour quotient in the responses based on the user profile; *(d)* ensure diversity of the seed utterances to increase user engagement and facilitate better communication; *(e)* frame the responses displaying positiveness and affirmativeness aiming to boost the user's morale. A few examples of seed utterances are given in Table 4 in Section A.1 of the Appendix.

**Dialogue Generation.** Once the seed utterances are finalized, the GPT-J model is utilized to generate new utterances. The overall dialogue generation completes in two stages. In the first stage, we manually design three prompts, each consisting of instruction and designed seed utterance. For each of the three prompts, 40 dialogues are created by feeding them to GPT-J. During generation, top-$k$ sampling (Fan et al., 2019; Radford et al., 2019) is employed, resulting in the generation of three candidate responses for each input utterance. To check alignment of the candidates with the ongoing conversation's context, the contextual similarity is computed between them using BERTScore-F1 ($BS_{F1}$) (Zhang et al., 2019). The candidate with $\max(BS_{F1})$ is selected as the final response. These generated dialogues are then manually rated by the same six human experts for quality on a Likert scale of 1-low, 2-moderate, and 3-high. An inter-evaluator Kappa (McHugh, 2012) agreement ratio of 76.3% is observed among the experts. The prompt generating the maximum number of dialogues with a score of 3 is selected as the final prompt. A prompt example is shown in Section A.1 of the Appendix. The remaining dialogues generated using selected prompt with a score of either 1 or 2 are manually corrected by the experts to ensure quality dialogues. In the second stage, the selected prompt along with the generated seed dialogues is given as input to the GPT-J model, which generates the dialogues with $n$ number of utterances in an incremental way. The dialogues with utterances having $BS_{F1} < 0.4$ are filtered out for manual cross-verification by human experts who are requested to correct the dialogues as per the guidelines.

### 3.1.2 Data Cleaning and Quality Control

After obtaining the entire conversational dataset, each dialogue is assessed in terms of *humanness*, *user profile consistency*, and *psychotherapeutic approach consistency* by the same set of experts. A score of 1-low, 2-moderate, or 3-high is assigned to each utterance. After obtaining the ratings, we observe an agreement ratio (McHugh, 2012) of 82.7%, 85.3% and 89.6% for *humanness*, *user profile consistency* and *psychotherapeutic approach consistency*, respectively among these experts. All dialogues having utterances with a score of 0 for any of these three aspects are discarded from the dataset. The final dataset statistics are given in Table 1.

| Metrics | Train | Validation | Test |
|---|---|---|---|
| *# of Dialogues* | 816 | 102 | 102 |
| *# of Utterances* | 19,568 | 2,692 | 2,811 |
| *Avg. Utterances per Dialogue* | 23.98 | 26.39 | 27.56 |

Table 1: Dataset statistics of PSYCON.

### 3.2 Dataset Annotation

To construct **e-THERAPIST**, we obtain the annotation for PSYCON dataset at two distinct levels-(i). dialogue-level annotation aspects, *viz.* user profile information- gender: *male (m)* and *female (f)*,

age: *young(y)*, *adult(a)*, and *elderly(e)*, persona: *openness (O)*, *conscientiousness (C)*, *extraversion (E)*, *agreeableness (A)* and *neuroticism (N)*, and psychotherapeutic approach: *directive (d)*, *non-directive (nd)* and *eclectic (ec)*[2]; (ii). utterance-level annotation aspects, *viz.* sentiment, politeness, and interpersonal behaviour. The annotation process involves the collaboration of three annotators, consisting of two Ph.D. holders in Linguistics and one individual with a Master's degree[3]. All three annotators possess excellent proficiency in English, substantial experience in labeling tasks, and a thorough understanding of sentiment, politeness, and interpersonal behavioural aspects.

### 3.2.1 Sentiment, Politeness, and Interpersonal Behaviour Annotation

The user's and therapist's utterances in PSYCON are annotated with one of the ternary sentiment labels, *viz. positive, negative* and *neutral*, and politeness labels *viz. polite, moderately_polite* and *impolite*, respectively.

The different interpersonal behaviour labels following the two-dimensional IPC model (Cremers et al., 2021) that we use in our work are: *directing (Dg), helpful (Hl), understanding (Ug), complaint (Ct), imposing (Ig), confrontational (Cl), dissatisfied (Dd)* and *uncertain (Un)*. This interpersonal behaviour annotation list has been extended to incorporate one more label, namely *empathetic (Em)*, considering the significance of empathy in therapy (Sharma et al., 2020, 2021; Saha et al., 2022). Due to space constraints, the description of all the nine interpersonal behaviour labels and the dataset annotation procedures are provided in Section A.1 of the Appendix.

## 4 e-THERAPIST

We warm start with a pre-trained language model GPT-2 medium (Radford et al., 2019) fine-tuned on a PSYCON denoted by $TC_N$. Each conversation in $TC_N$ can be represented by $TC_k = \{t_0, u_0, .., t_i, u_i, .., t_{T-1}, u_{T-1}\}$; $t_i$ and $u_i$ give the therapist's and user's $i^{th}$ utterance in the conversation, respectively, where $0 <= k < N$ for $N$ number of conversations. For each $TC_k$, the user's corresponding gender, age, and persona can be represented by $g_k = \{m, f\}$, $a_k = \{y, a, e\}$

---

[2]The dialogue-level information is obtained during Data Cleaning and Quality Control stage described in Section 3.1.2.

[3]The annotators are compensated according to institutional guidelines.

and $p_k = \{O, C, E, A, N\}$, respectively. Further, for each user utterance, $u_i$ in $TC_k$, corresponding sentiment $s_i = \{negative, neutral, positive\}$ is predicted by sentiment classifier $SC$. The concatenated $u_i$ with $s_i$ can be given as $u_i^s = [u_i + s_i]$. Now, the $lm_\theta$ is trained to predict output $y_i \approx t_i$ given input $x_i = [u_i^s + u_{i-1}^s + t_{i-1}] + [g_k + a_k + p_k]$ in a supervised learning setting. It can be given as:

$$lm_\theta(TC_n) = \prod_{k=0}^{N} \prod_{i=0}^{i=T} \rho(y_i | x_i) \qquad (1)$$

We call the trained $lm_\theta$ as supervised learning fine-tuned language model (SLLM). Next, to end up with proposed **e-THERAPIST**, we fine-tune SLLM using our novel reward model in RL setting to generate user's profile and sentiment-aware polite responses with correct interpersonal behaviour in therapeutic conversations. The architecture of the proposed system is shown in Figure 2.

### 4.1 Reward Model

For a given input $x_i$, we generate a set of $n$-$y_i$ possible candidates using $lm_\theta$ and score them using a reward model. Our reward model consists of seven distinct rewards to adapt correct polite interpersonal behaviour in generated responses of $lm_\theta$. First reward $R_1$ focuses on the awareness of the user's gender and age. Secondly, $R_2$ emphasizes the awareness of the user's persona. $R_3$ drives the model to follow the correct psychotherapeutic approach. $R_4$, ensures appropriate politeness as per the user's sentiment. $R_5$ aims to adapt the interpersonal behaviour in accordance with the user's sentiment. $R_6$ focuses on maintaining the plausibility of the context and individual utterances. Lastly, $R_7$ steers the generated responses to be fluent and diverse.

All seven rewards can be categorized into two types of rewards, *viz. Attribute-specific* Rewards ($R_1$, $R_2$, $R_3$, $R_4$, $R_5$) - to reinforce different attributes of the user or therapist in generated responses, and *Response-quality* Rewards ($R_6$, $R_7$) - to enforce contextually fluent and adequate responses in a conversation.

### 4.1.1 Attribute-specific Rewards

To design each of the $R_1$, $R_2$, $R_3$, $R_4$ and $R_5$, five different RoBERTa-large (Liu et al., 2019) based classifiers are fine-tuned. For $R_1$, Gender-Age Classifier ($GAC$) - predicts one of the six classes of gender and age $GAC(t_i) = \{fy, fa, fe, my, ma, me\}$,

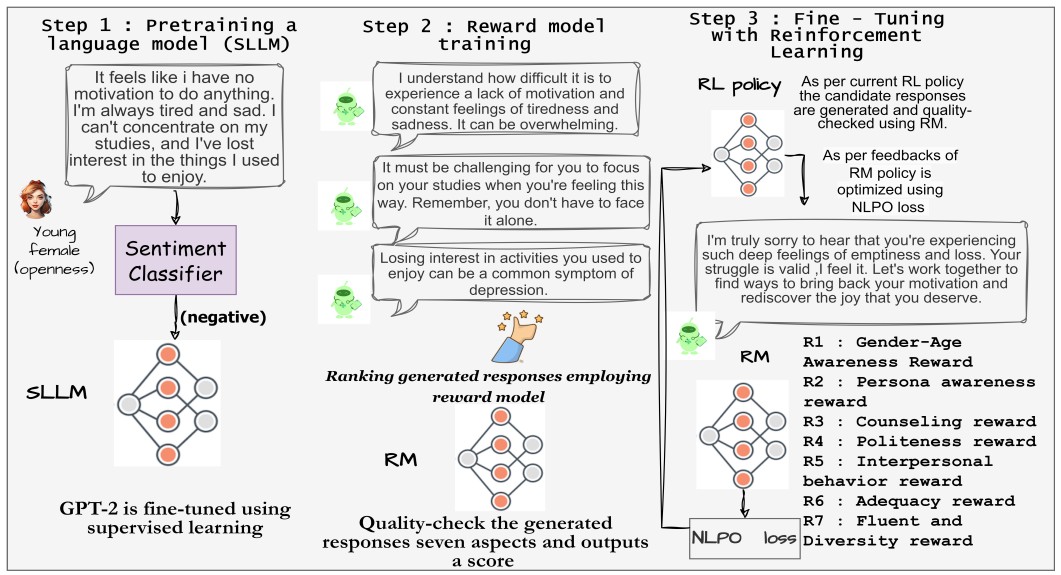

Figure 2: Overall architecture of the proposed system

where $fy, fa, fe, my, ma, me$ denote female-young, female-adult, female-elder, male-young, male-adult, and male-elder, respectively. For $R_2$, Persona classifier ($PC$) - predicts one of the five personality traits $PC(t_i) = \{O, C, E, A, N\}$. For $R_3$, a psychotherapeutic approach classifier $CTC([t_i + u_i]) = \{d, nd, ec\}$. For $R_4$, a sentiment classifier $SC(u_i) = \{positive, neutral, negative\}$ predicts the sentiment of the user's utterance and a politeness classifier $PoC(t_i) = \{impolite, moderately\_polite, polite\}$ predicts the politeness of the therapist's utterance. Lastly, for $R_5$, an interpersonal behaviour classifier $IBC(t_i) = \{Dg, Hl, Ug, Ct, Ig, Cl, Dd, Un, Em\}$ predicts one of the nine interpersonal behaviour labels[4]. To design each reward, we track the true class probabilities score from each of the classifiers. $R_1$ and $R_2$ penalize those responses which deviates from true user profiles *viz.* gender, age and persona and are computed as:

$$R1 = GAC(t_i) - \alpha \times GAC(y_i) \quad (2)$$

$$R2 = PC(t_i) - \alpha \times PC(y_i) \quad (3)$$

$R_3$ penalize the responses deviating from the correct psychotherapeutic approach in the ongoing dialogue:

$$R3 = CTC([t_i + u_i]) - \alpha \times CTC([y_i + u_i]) \quad (4)$$

---

[4]The accuracies of $GAC$, $PC$, $SC$, $PoC$, $CTC$, and $IBC$ are 89.4%, 89.7%, 91.2%, 95.6%, 92.3%, and 89.8%, respectively.

To formulate $R_4$, and $R_5$, we penalize the responses that do not adapt true politeness and interpersonal behaviour as per the sentiment of the user.

$$R4 = PoC(t_i + SC(u_i)) - \alpha \times PoC(y_i + SC(u_i)) \quad (5)$$

$$R5 = IBC(t_i + SC(u_i)) - \alpha \times IBC(y_i + SC(u_i)) \quad (6)$$

$\alpha$ acts as a penalization factor in all of the rewards $R_1$, $R_2$, $R_3$, $R_4$, and $R_5$[5]. The final attribute-specific reward can be written as: $R_A = \beta_1 R_1 + \beta_2 R_2 + \beta_3 R_3 + \beta_4 R_4 + \beta_5 R_5$[6].

### 4.1.2 Response-quality Rewards

In $R_6$, the candidates deviating from context and user's utterance are penalized. It is computed using BERTScore-F1 ($BS_{F1}$) (Zhang et al., 2020) between (i) the true context input $x_i$ and generated $y_i$ (ii) user's utterance $u_i$ and generated $y_i$. To avoid rewarding high similarities steering, a threshold of 1 is taken into account.

$$R_6 = \frac{min((BS_{F1}(x_i, y_i) + BS_{F1}(u_i, y_i)), 1)}{2} \quad (7)$$

$R_7$ ensures fluency and non-repetitiveness in the generated responses and is computed as the sum of the perplexity reciprocal and $BS_{F1}$ between generated $y_i$ and $y_{i-1}$.

$$R_7 = \frac{1}{PPL} + BS_{F1}(y_i, y_{i-1}) \quad (8)$$

---

[5]The value of $\alpha$ is taken as greater than or equal to 1.
[6]$\beta_1 + \beta_2 + \beta_3 + \beta_4 + \beta_5 = 1$

The response-quality reward can be written as: $R_R = \gamma_1 R_6 + \gamma_2 R_7$[7]. Lastly, by combining $R_A$ and $R_R$, we obtain our final normalized reward function $\hat{R}$ as follows:

$$\hat{R} = \frac{\delta_1 R_A + \delta_2 R_R}{7} \quad (9)$$

The final score $\hat{R}$ is utilized in an RL policy loss NLPO (Ramamurthy et al., 2022) to provide feedback, compelling the agent to generate high-quality candidates that align with the preferred outcomes. During RL-fine tuning, RL-policy is initialized with $\pi_\theta = lm_\theta$. NLPO basically implements a masking policy $\pi_\phi$, a copy of the current policy $\pi_\theta$ updated after $m$ steps. Top-$p$ tokens in a vocabulary $V$ contribute as valid-mask remaining are subjected to an invalid mask, i.e. selection probability is set to zero for these tokens. Hence, this strikes a balance between incorporating more task-relevant information compared to the KL penalty derived from $\pi_\theta$. Due to space restrictions, details of NLPO are given in Section A.2 of the Appendix.

## 5 Experiments

**Baselines.** We compare the performance of proposed **e-THERAPIST** with seven strong baselines - LM (Radford et al., 2019), ARDM (Wu et al., 2021b): LMs trained alternatively for both user and therapist, GPT-Critic (Jang et al., 2022): improving LM through cloning of critic-guided self-generated sentences during fine-tuning, SLLM: Fine-tuned LM in a supervised setting with user profiles and sentiment, SLLM+PPO: Supervised + PPO loss based fine-tuning, e-THERAPIST-R: **e-THERAPIST** with $\hat{R} = 0$, e-THERAPIST-ASR: **e-THERAPIST** with $R_A = 0$ and e-THERAPIST-RQR: **e-THERAPIST** with $R_R = 0$.

**Implementation.** We experiment with different values of $n = \{2, 3, 4, 5, 8\}$, and found that $n = 3$ yields the best performance. We use GPT-2 as the language model (LM) and context window size of 4. We employ top-k sampling with k = 20 as the decoding method for all the models in our work. For RL-based models, we determine the end values of coefficients $\beta_1 = 0.1$, $\beta_2 = 0.2$, $\beta_3 = 0.2$, $\beta_4 = 0.2$, $\beta_5 = 0.3$, $\gamma_1 = 0.5$ $\gamma_2 = 0.5$, $\delta_1 = 0.75$, and $\delta_2 = 0.25$ empirically. Due to space restrictions, we have included detailed implementations and weight optimization in sections A.3 and A.5 of the Appendix, respectively.

---
[7] $\gamma_1 + \gamma_2 = 1$

**Evaluation Metrics.** To evaluate the performance of the proposed system, **e-THERAPIST**, both automatic and human evaluation are conducted. All the five classifiers, *viz. GAC*, *PC*, *CTC*, *PoC*, and *IBC* are evaluated in terms of Weighted Accuracy (W-ACC) and Macro-F1. In automatic evaluation, **e-THERAPIST** is evaluated w.r.t (i). attribute-success - Gender-Age consistency ($GA_c$), Persona consistency ($P_c$), Psychotherapeutic approach correctness ($CT_c$), Politeness correctness ($Po_c$) and Interpersonal behaviour correctness ($IB_c$)[8], and (ii). language-quality - Perplexity ($PPL$), BERTScore-F1 ($BS_{F1}$), Response-length ($R\_LEN$).

For human evaluation, we ask the same six experts to evaluate the **e-THERAPIST**. Initially, each evaluator engages with the system 5 times, with a different set of responses each time. These 5 human-evaluated interactions are then cross-verified by psychotherapists from a government-run institution to ensure evaluation quality. Upon passing verification, the additional 90 interactions (15 per evaluator) are evaluated, resulting in a total of 120 human-evaluated dialogues. Human evaluation metrics also includes $GA_c$, $P_c$, $CT_c$, $Po_c$, $IB_c$ as attribute-success and Fluency ($F$), Consistency ($C$), and Non-Repetitiveness ($N_R$). All dialogue interactions are evaluated on an integer Likert scale of 1-5[9] [10].

## 6 Results and Analysis

**Automatic Evaluation.** Table 2 depicts that the proposed **e-THERAPIST** achieves better results w.r.t. all the eight baselines, *viz.* LM, ARDM, GPT-Critic, SLLM, SLLM+PPO, e-THERAPIST-R, e-THERAPIST-ASR, and e-THERAPIST-RQR in terms of all the seven metrics, *viz.* $GA_c$, $P_c$, $CT_c$, $Po_c$, $IB_c$, $PPL$, $BS_{F1}$, and $R_{LEN}$. Better results of SLLM compared to LM, ARDM, and RL-based GPT-critic highlight the importance of user's gender, age and persona profiles. Incorporation of these attributes steer the SLLM towards more interactive responses inherently. e-THERAPIST-R performs comparable to SLLM as in the absence of any reward, RL policy acts $\pi_\theta \approx lm_\theta$.

It can also be observed that in the absence of only attribute-specific rewards, the performance

---
[8] $GA_c$, $P_c$, $CT_c$, $Po_c$, $IB_c$ are computed by five respective classifiers with accuracies.

[9] 1-5 denotes low to high.

[10] An inter-evaluator agreement ratio of 72.3% is observed in evaluations.

| Model | $GA_c$ | $P_c$ | $CT_c$ | $Po_c$ | $IB_c$ | $PPL$ | $BS_{F1}$ | $R_{LEN}$ |
|---|---|---|---|---|---|---|---|---|
| LM (Radford et al., 2019) | 78.4% | 72.1% | 79.5% | 80.2% | 73.6% | 4.26 | 0.68 | 15.61 |
| ARDM (Wu et al., 2021b) | 80.4% | 73.3% | 80.0% | 81.5% | 74.2% | 3.57 | 0.74 | 16.82 |
| GPT-Critic (Jang et al., 2022) | 80.7% | 73.8% | 80.6% | 82.7% | 73.1% | 3.86 | 0.69 | 15.94 |
| SLLM | 85.4% | 80.1% | 86.3% | 84.6% | 77.8% | 3.26 | 0.81 | 19.79 |
| e-THERAPIST-R | 85.1% | 79.7% | 86.8% | 84.5% | 77.5% | 3.09 | 0.84 | 19.26 |
| e-THERAPIST-ASR | 86.1% | 80.8% | 87.2% | 86.2% | 79.8% | 3.06 | 0.87 | 20.12 |
| e-THERAPIST-RQR | 87.5% | 82.3% | 88.7% | 87.9% | 80.5% | 2.97 | 0.88 | 22.79 |
| SLLM+PPO | 89% | 83.9% | 91.5% | 91.3% | 82.3% | 2.67 | 0.89 | 23.01 |
| **e-THERAPIST** | **90.1%** | **84.1%** | **92.6%** | **92.5%** | **83.4%** | **2.52** | **0.89** | **23.89** |

Table 2: Automatic evaluation results

| Model | $GA_c$ | $P_c$ | $CT_c$ | $Po_c$ | $IB_c$ | $F$ | $C$ | $N_R$ |
|---|---|---|---|---|---|---|---|---|
| LM (Radford et al., 2019) | 2.02 | 2.21 | 2.07 | 2.10 | 2.39 | 2.17 | 2.39 | 2.01 |
| ARDM (Wu et al., 2021b) | 2.88 | 2.74 | 2.77 | 2.81 | 2.80 | 2.79 | 2.83 | 2.29 |
| GPT-Critic (Jang et al., 2022) | 2.98 | 2.83 | 2.81 | 2.90 | 2.91 | 2.86 | 2.91 | 2.34 |
| SLLM | 3.50 | 3.67 | 3.80 | 3.75 | 3.41 | 3.89 | 3.44 | 3.21 |
| e-THERAPIST-R | 3.53 | 3.45 | 3.86 | 3.84 | 3.50 | 4.11 | 4.05 | 3.72 |
| e-THERAPIST-ASR | 3.75 | 3.70 | 4.01 | 3.91 | 3.72 | 4.32 | 4.27 | 3.82 |
| e-THERAPIST-RQR | 3.97 | 3.91 | 4.12 | 4.09 | 3.89 | 4.45 | 4.33 | 3.97 |
| SLLM+PPO | 4.10 | 4.06 | 4.38 | 4.30 | 4.01 | 4.55 | 4.50 | 4.05 |
| **e-THERAPIST** | **4.21** | **4.10** | **4.42** | **4.35** | **4.02** | **4.62** | **4.60** | **4.08** |

Table 3: Human evaluation results.

of e-THERAPIST-ASR sees a significant increase in scores of $PPL$, $BS_{F1}$, and $R_{LEN}$, but minimal change is attribute-specific metrics. This supports the use of Response-quality rewards, due to which model tries to engage the user with longer and contextually-adequate responses. Similarly, an increase in attribute-specific metrics in case of e-THERAPIST-RQR supports the requirement of attribute-specific rewards as well. It can also be inferred from both e-THERAPIST-ASR and e-THERAPIST-RQR results that the presence of each type of rewards affects other metrics positively but with a minimal margin. This means that all rewards interact with each other and helps in achieving user-aware polite interpersonal therapy. SLLM+PPO is same as **e-THERAPIST** only with the difference of loss. In the performance of both the models, there is little margin, but still, it reflects that NLPO drives the model to generate longer responses with correct incorporation of politeness, interpersonal behaviour and psychotherapeutic approach. Automatic evaluations support our hypothesis that knowledge of user's profile and reward modelling can play a significant role in building a better psychotherapy dialogue system.

**Human Evaluation.** Results of human evaluation for **e-THERAPIST** are obtained in sync with the automatic evaluation metrics as shown in Table 3. For all the metrics, *viz.* $GA_c$, $P_c$, $CT_c$, $Po_c$, $IB_c$, $F$, $C$, and $N_R$, **e-THERAPIST** achieves better scores as compared to all the eight base-

lines. Here, as well, SLLM and e-THERAPIST-R beat LM, ARDM, and GPT-Critic. Further, e-THERAPIST-ASR shows a small marginal increase on SLLM and e-THERAPIST-R. This implies that response-quality rewards are not enough to ensure user's profile and sentiment-aware therapeutic responses. Notably, difference between performance of THERAPIST-RQR and of SLLM+PPO is of small margin. Similarly, minimal performance difference can be seen between SLLM+PPO and **e-THERAPIST** as well. This leads to the argument that attribute-specific rewards play a crucial role in adapting correct psychotherapeutic approach and interpersonal behaviour in generated responses while imbuing politeness in them. It should also be pointed out that **e-THERAPIST** achieves much better scores w.r.t. all metrics as compared to THERAPIST-RQR. This highlights the importance of response-quality rewards as well. Thus, it can be concluded that a robust psychotherapeutic dialogue system can be built by striking a balance between attribute-specific and response-quality rewards. Employing both, the user can be engaged in interpersonal therapeutic communication while also maintaining a cordial environment.

## 7 Conclusion

This work introduces **e-THERAPIST**, a novel polite interpersonal dialogue system for psychotherapy. First, considering user profile information (gender, age, persona) and the therapist's psy-

chotherapeutic approach, a new conversational dataset PSYCON is created by prompting GPT-J with manual interventions. Further, it is annotated with user's sentiment and therapist politeness label and interpersonal behaviour. Then, a psychotherapeutic dialogue system **e-THERAPIST** is developed in an RL framework. To ensure the preferences of gender, age, persona and sentiment-aware polite interpersonal psychotherapeutic responses, a novel reward function is designed consisting of seven rewards. Our results concludes the requirement of each of the reward to ensure polite and interpersonal psychotherapeutic responses tailored to user's profile and attributes, and eventually contributing to improved therapy experiences. In future, we would like to incorporate external knowledge to facilitate more factual conversations.

## Limitations

**e-THERAPIST** comes with some limitations. Firstly, the training process requires a substantial GPU memory capacity i.e. of 40 GB. Another challenge arises from the optimization of reward weights, which can significantly extend the training and validation time. To address this, heuristic approaches are used to select specific combinations of reward weights. It is also observed, in the case of continuous, short, or direct responses (e.g., *'Yes'*, *'I don't know'*, *'No'*, *'2'*, *'Yeah'*) model initially attempts to provide therapy by inquiring about the user's issue; however, after a few turns, the model may generate repetitive or inconsistent responses. This is because the training data primarily consists of interactive dialogues with longer utterances, leading to confusion in handling concise inputs. It is also observed that continuous out-of-the-context responses may drive the model towards generation of inadequate responses, as these large language models models inherently have knowledge of vast information in their memories.

To overcome these limitations, future research could focus on refining and enhancing a psychotherapeutic dialogue systems as well as language models. This could involve addressing memory requirements, reducing training time, improving the handling of short, direct and out-of-context responses, and ensuring the generation of relevant inquiries to enhance user satisfaction.

## Ethics Statement

Developing a psychotherapeutic dialogue system do take a step towards Responsible AI. But, same methodology can also be used to develop toxic or irresponsible natural language generation models. Hence, we acknowledge that the significant importance of addressing ethical concerns when developing computational models for psychotherapeutic applications is needed. Given the sensitive nature of the subject, we prioritize safeguarding the privacy of users' personal data. It is important to clarify that this paper does not provide any clinical diagnostic assertions. We mainly focus to enrich the interactiveness of such dialogue system with the user better engagingness and therapy dialogues. The dataset will be made available for research purposes with proper permissions.

## Acknowledgements

Kshtij Mishra acknowledges the support received from the Indian government's "Prime Minister's Research Fellowship (PMRF) Program". Priyanshu Priya acknowledges the financial support provided by the Department of Science and Technology, Ministry of Science and Technology, Government of India, through the Innovation in Science Pursuit for Inspired Research (INSPIRE) Fellowship.

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

# A Appendix

## A.1 Dataset

We provide the details of the dataset in this section.

### A.1.1 Prompt Design

```
begin
    psychological Issue: depression
    gender: female
    age: young
    persona: openness
    psychotherapeutic approach: directive Technique
    dialogue flow: problem description, self-care, coping strategies
    topic: "problem description"
        User: Hi, I've been struggling with depression lately, and I'm
        seeking counseling to help me cope with it.
    topic: "problem description→self-care"
        Agent: I'm here to assist you. It's great that you've
        recognized the need for support. In addition to counseling,
        incorporating self-care practices into your daily routine can be
        beneficial for managing depression. Have you tried any self-
        care activities before?
    topic: "self-care"
        User: I've dabbled in self-care, but I'm looking for more
        guidance on effective strategies to improve my mental well-
        being.
    topic: "self-care"
        Agent:
end
```

Figure 3: Example of one-shot version of prompt.

The proposed PSYCON dataset comprises conversations focusing on seven psychological issues and different gender, age, personas and psychotherapeutic approach. In order to create synthetic data focusing on these aspects, we prompt the GPT-J model utilizing four-shot prompt. A sample of the one-shot version of the prompt is provided in Figure 3. We provide the psychological issue, gender, age, persona, psychotherapeutic approach, dialogue flow and the seed utterances for which the next utterance needs to be generated. Table 4 depicts a few examples of seed utterances. The four-shot prompt adheres to a similar pattern with four examples in the input sequence. A few turns of a dialogue generated using this method are given in Table 6.

### A.1.2 Dataset Annotation Details

The entire annotation process for sentiment, politeness and interpersonal behaviour labels proceeds in

two stages to reduce manual efforts. First, we randomly sample 340 dialogues from the dataset and then ask all three annotators to manually annotate the user's utterances with the required sentiment label and the therapist's utterance with the relevant politeness and interpersonal behaviour labels. Second, three pre-trained RoBERTa-large (Liu et al., 2019) models are fine-tuned on manually annotated samples to build sentiment, politeness, and interpersonal behaviour classifiers. Then, the remaining utterances are passed through the respective classifier to predict the corresponding label. Lastly, the same annotators are asked to cross-verify the predicted labels and correct them, if needed. A reliable multi-rater Kappa (McHugh, 2012) agreement ratios of 85.2%, 77.1% and 73.6% is observed in the first stage and 89.6%, 84.6% and 80.3% is observed in the second stage for sentiment, politeness and interpersonal behaviour labels, respectively.

A sample conversation with the annotation for sentiment, politeness and interpersonal behaviour is shown in Figure 4. The description of different interpersonal behaviour labels along with examples are provided in Table 5.

## A.2 NLPO

The RL-based parameterized control policy for e-THERAPIST can be defined as:

$$\pi_\theta : S \rightarrow \Delta(A) \tag{10}$$

Here, $\pi_\theta$ is a probability function that attempts to select an action $A$ in a given state space $S$ with a goal to maximize long-term discounted rewards $R$ over a trajectory:

$$E_\pi[P(\sum_{t=0}^{T} \gamma_t R(s_t, a_t)] \tag{11}$$

. During RL fine-tuning, we initialize $\pi_\theta = lm_\theta$ The value function $V^\pi$ and Q-value function $Q^\pi$ for policy $\pi_\theta$ and estimated reward $R$ are be computed as:

$$V_t^\pi = E_{a_t} \sim \pi \left[ \sum_{\tau=t}^{T} \gamma R(s_\tau, a_\tau, y) \right] \tag{12}$$

$$Q_t^\pi(s_t, a_t) = R(s_t, a_t, y) + \gamma E_{s_{t+1} \sim P}[V_{t+1}^\pi(s_{t+1})] \tag{13}$$

Considering $V^\pi$ and Q-value function $Q^\pi$, we compute the advantage estimate $A^\pi$ as:

$$A_t^\pi(s, a) = Q_t^\pi(s, a) - V_t^\pi \tag{14}$$

. To stabilize the training, we approximate the advantage using Generalized Advantage Estimation(Schulman et al., 2015). To address the sparsity of sequence-level rewards in the environment, we apply Then, a regularization (Wu et al., 2021a) is applied address the sparsity of sequence-level rewards in the environment. It basically incorporates a token-level KL penalty into the reward function to discourage the model from deviating too far from the last model $\pi_\theta$. It can be formalized as:

$$\hat{R}(s_t, a_t, y) = R(s_t, a_t, y) - \delta KL(\pi_\theta(a_t|s_t)||lm_\theta(a_t|s_t)) \tag{15}$$

Here, $\hat{R}$ represents the regularized KL reward, y denotes the ground-truth predictions, $KL(\pi_\theta(a_t|s_t)||lm_\theta(a_t|s_t))$ is the KL divergence between $(\pi_\theta(a_t|s_t)$ and $lm_\theta(a_t|s_t))$. The KL coefficient $\delta$ is dynamically adjusted following (Ziegler et al., 2019). Now, we update the RL policy $\pi_\theta$ by maximizing the PPO-Clip objective:

$$\pi_{\theta_{m+1}} = \text{argmax}_\theta \frac{1}{|D_m|T}$$

$$\sum_{\tau \in D_m} \sum_{t=0}^{T} \min(r_t(\theta) A^{\pi_{\theta_m}}(\tau_t), \text{clip}(r_t(\theta),$$

$$1 - \epsilon, 1 + \epsilon) A_{\pi_{\theta_m}}(\tau_t)$$

where $r_t(\theta) = \frac{\pi_\theta(a_t|s_t)}{\pi_{\theta_m}(a_t|s_t)}$. Next, value function is updated as follows:

$$V_{\phi_{m+1}} = \text{argmin}_\phi \frac{1}{|D_m|} \sum_{\tau \in D_m} \sum_{t=0}^{T} \left( V_\phi(s_t) - \hat{R}_t \right)^2 \tag{16}$$

. For every $m$ iterations, the parameterized masked policy is also updated as follows:

$$\pi_\psi^{n+1}(\cdot|\cdot, \pi_{\theta m+1}) \tag{17}$$

.

## A.3 Implementation Details

All classifiers are fine-tuned using the RoBERTa-large (Liu et al., 2019). The language models GPT-2-medium (Radford et al., 2019), ARDM (Wu et al., 2021b), and SLLM are trained with a cross-entropy loss. In supervised learning setting, AdamW optimizer(Loshchilov and Hutter, 2018) is empoloyed with a learning rate of $\alpha = 2e^{-05}$ and $seed\_value = 10$.

For SLLM+PPO and **e-THERAPIST**, training is conducted with $batch\_size = 8$, $seed\_value = 10$, $human\_reward = 10$, $max\_candidate\_length = 50$, $clip\_ratio = 0.2$, $discount\_factor = 0.95$, $number\_of\_steps = 32000$, $steps\_per\_update = 640$ and *AdamW* optimizer (Loshchilov and Hutter, 2018) with a learning rate of $\alpha = 2e^{-05}$, $\varepsilon = 0.2$ and $epochs = 20$.

| | Seed Utterances |
|---|---|
| Psychological Issue: PTSD
Gender: Male
Age: Young
Persona: Openness
Psychotherapeutic Approach: Eclectic | *User*: I've been struggling with PTSD lately. It's been really tough to cope with. |
| | *Therapist*: As a therapist, I understand that PTSD can have a significant impact on your life.
I'm here to help you navigate through it. |
| | *User*: I'm a young male dealing with PTSD, and I'm looking for effective techniques to manage it.
Can you suggest any approaches? |
| Psychological Issue: Depression
Gender: Female
Age: Adult
Persona: Conscientiousness
Psychotherapeutic Approach: Directive | *User*: I've been feeling really down lately. I think I might be experiencing depression. |
| | *Therapist*: I understand how challenging depression can be.
I'm here to provide support and guidance. |
| | *User*: I'm an adult female struggling with depression, and I've heard about the directive technique.
Can you tell me more about it and how it could help me? |
| Psychological Issue: Stress
Gender: Female
Age: Elder
Persona: Extraversion
Psychotherapeutic Approach: Directive | *User*: I think a lot of my stress comes from feeling overwhelmed by all my responsibilities. |
| | *Therapist*: I understand. Stress can affect us in various ways.
Can you tell me more about what's been going on? |
| | *User*: I've always been more on the extraverted side, so being isolated during the pandemic has been
particularly challenging for me. I'm used to being active and engaged with others. |
| Psychological Issue: PTSD
Gender: Male
Age: Adult
Persona: Agreeableness
Psychotherapeutic Approach: Non-directive | *User*: It's been going on for several months now. The intensity varies, but there are moments
when I feel completely overwhelmed by these memories. |
| | *Therapist*: I can understand how distressing and disruptive these symptoms can be.
Have you sought any professional help or support for your PTSD symptoms? |
| | *User*: No, I haven't. I've been hesitant to reach out because I've always tried to handle things on my own. |
| Psychological Issue: Anxiety
Gender: Male
Age: Young
Persona: Neuroticism
Psychotherapeutic Approach: Non-directive | *User*: Lately, I've been feeling overwhelmed by anxiety. |
| | *Therapist*: Thank you for sharing that with me. Anxiety can be quite overwhelming.
Can you describe what you're specifically experiencing? |
| | *User*: I've always been someone who tends to be more neurotic, so this anxiety is really overwhelming. |

Table 4: A few examples of seed utterances.

| Interpersonal behaviour | Definition | Example |
|---|---|---|
| *Directing* | Provides clear instructions, assistance and guidance to the users. | *Okay, well let's start with something simple. Maybe you can try going for a walk outside every day.* |
| *Helpful* | Active listening, providing reassurance and emotional support to the users based on their circumstances in order to motivate them. | *One thing you can try is practising self-compassion to fight depression. That means treating yourself with the same kindness and understanding that you would offer to a friend.* |
| *Empathetic* | Conveys a sense of being heard, valued and validated to create a non-judgmental and caring environment so as to understand and solve the user's problem. | *That sounds really tough. Have you been experiencing these feelings for a long time?* |
| *Understanding* | Reflects the ability to acknowledge and accept others' perspectives. | *It's completely understandable to have reservations. Take your time and consider joining support groups at your own pace.* |
| *Compliant* | Reflects the sense of accommodation, cooperation and confirmation with the expectations and demands of users. | *I understand that you're feeling upset about the way your friend treated you. Let's explore strategies to improve your communication and set boundaries in a healthy way.* |
| *Imposing* | Reflects an attempt to assert control over the user's emotions, thoughts, and experiences for their betterment. | *It can be challenging, but it's a skill that can be developed over time. I can recommend some exercises you can try to help you get started.* |
| *Confrontational* | Reflects a direct or assertive approach that confronts or challenges the user's thoughts and emotions to make users understand their self-defeating patterns or manipulations. | *That sounds really challenging. It's important to remember that your worth isn't tied to your job performance. Have you considered talking to your boss about how you're feeling?* |
| *Dissatisfied* | Reflects a usual feeling of unhappiness or discontentment with the user's behaviour or actions. Identifying the source of dissatisfaction facilitates making quicker progress towards the user's goals. | *Perhaps, if you feel comfortable, we can explore that topic further during our next session.* |
| *Uncertain* | Reflects the lack of clarity about the user's perspectives and experiences. Recognizing the source of uncertainty fosters healthy and supportive interactions. | *I'm not quite sure, but would it be helpful if we explored that topic further together?* |

Table 5: The definition and example of different interpersonal behaviour in our annotated dataset.

### A.3.1 Hardware Configuration

The experimental setup includes the following device configurations: (a) GPU: A100-PCIE-40GB, (b) CUDA Support: CUDA 11.x (or later.) (c) GPU clocks: Base: 765 MHz, Boost: 1410 MHz, (d) Memory Size: 40 GB, (e) Memory Type: HBM2, (f) Memory clock: 1215 MHz, (g) Bus Width: 5120 bits, (h) Total board power: 250 W.

| Psychological Issue | Post-traumatic stress disorder |
|---|---|
| Gender | Male |
| Age | Young |
| Persona | Openness |
| Psychotherapeutic Approach | Eclectic |
| Seed Utterances | *User*: Hi, I've been dealing with PTSD recently and it has been quite challenging. |
| | *Therapist*: Hi, I'm sorry to hear that you've been dealing with PTSD. Could you provide me with more details about your experience? |
| | *User*: Of course. I've been grappling with PTSD, and I'm uncertain about effective coping strategies. |
| **Speaker** | **Generated Utterances** |
| *Therapist* | Hello, it's nice to meet you. How can I help you today? |
| *User* | Hi, I've been struggling with PTSD and I'm not sure how to cope with it. |
| *Therapist* | I'm sorry to hear that. Would you mind telling me more about your experiences with PTSD? |
| *User* | It started after I witnessed a traumatic event a few years ago. Ever since then, I've had trouble sleeping and have intense flashbacks. |
| *Therapist* | That sounds really difficult. Have you tried any coping mechanisms in the past? |
| *User* | I've tried some breathing exercises and meditation, but they don't seem to help much. |

Table 6: A sample dialogue from PSYCON generated using our method.

## A.4 Classifier Results

Results of all five classifiers are shown in Table 7. From this table, it can be inferred that RoBERTa-large based classifiers achieve better scores for **W-ACC** and **Macro-F1** as compared to the BERT-large based classifiers.

| | BERT-large | | RoBERT-large | |
|---|---|---|---|---|
| **Classifier** | **W-ACC** | **Macro-F1** | **W-ACC** | **Macro-F1** |
| GAC | 0.882 | 0.871 | 0.894 | 0.873 |
| PC | 0.873 | 0.860 | 0.897 | 0.875 |
| SC | 0.893 | 0.881 | 0.912 | 0.904 |
| CTC | 0.904 | 0.893 | 0.923 | 0.921 |
| PoC | 0.947 | 0.942 | 0.956 | 0.943 |
| IBC | 0.872 | 0.752 | 0.898 | 0.872 |

Table 7: Evaluation results of the classifiers.

## A.5 Weight Optimization

To determine the optimal combination of weights for the reward function, we conducted experiments with different weight combinations of $\beta_1, \beta_2, \beta_3, \beta_4, \beta_5, \gamma_1, \gamma_2, \delta_1$, and $\delta_2$. These weights were validated using a 20% hold-out dataset PSYCON, and the combination that resulted in the highest perplexity score was selected for training **e-THERAPIST**. Table 8 presents the weights considered for optimization using the PSYCON dataset. The table indicates that considering all the rewards leads to a better perplexity score. Additionally, removing any of the rewards results in a decrease in the perplexity score, highlighting the importance of each reward in the model.

| WEIGHT OPTIMIZATION | | | | | | | | | |
| --- | --- | --- | --- | --- | --- | --- | --- | --- | --- |
| $\beta_1$ | $\beta_2$ | $\beta_3$ | $\beta_4$ | $\beta_5$ | $\gamma_1$ | $\gamma_2$ | $\delta_1$ | $\delta_2$ | **PPL** |
| 0.1 | 0.2 | 0.2 | 0.3 | 0.2 | 0.5 | 0.5 | 0.5 | 0.5 | 2.5261 |
| 0 | 0 | 0 | 0.8 | 0.2 | 0.55 | 0.45 | 0.4 | 0.6 | 2.556 |
| 0.1 | 0.1 | 0.1 | 0.7 | 0 | 0.1 | 0.9 | 0.9 | 0.1 | 2.6154 |
| 0 | 0 | 0 | 0 | 0 | 0 | 0 | 0 | 0 | 3.0963 |
| 0.1 | 0.2 | 0.2 | 0.2 | 0.3 | 0.35 | 0.65 | 0.45 | 0.55 | 2.5290 |
| 0.1 | 0.2 | 0.2 | 0.3 | 0.2 | 0.7 | 0.3 | 0.3 | 0.7 | 2.5361 |
| 0.2 | 0.1 | 0.1 | 0.4 | 0.2 | 0.8 | 0.2 | 0.2 | 0.8 | 0 2.5541 |
| 0.15 | 0.15 | 0.3 | 0.3 | 0.1 | 0 | 1 | 0.25 | 0.75 | 2.5961 |
| 0.3 | 0.3 | 0.3 | 0.1 | 0 | 0 | 1 | 1 | 0 | 2.6182 |
| 0.2 | 0.2 | 0.2 | 0.2 | 0.2 | 0.5 | 0.5 | 0.5 | 0.5 | 2.58932 |
| **0.1** | **0.2** | **0.2** | **0.2** | **0.3** | **0.5** | **0.5** | **0.75** | **0.25** | **2.52131** |

Table 8: Weight optimization using different reward weight combinations.

Figure 4: An example dialogue between the therapist (**T**) and the user (**U**) from PSYCON. The text highlighted in purple denotes the sentiment label of the user's utterances. The text highlighted in pink denotes the interpersonal behaviour and politeness labels of the therapist's utterances.