# OpenReview forum: "e-THERAPIST: I suggest you to cultivate a mindset of positivity and nurture uplifting thoughts"
_EMNLP/2023/Conference — EMNLP 2023 Main_

### Official Review · Reviewer_MZbm · 2023-08-01

**Soundness:** 3

**Ethical Concerns:**

Yes

**Excitement:**

2: Mediocre: This paper makes marginal contributions (vs non-contemporaneous work), so I would rather not see it in the conference.

**Missing References:**

[1] Towards Empathetic Open-domain Conversation Models: a New Benchmark and Dataset
[2] Towards Emotional Support Dialog Systems
[3] AugESC: Dialogue Augmentation with Large Language Models for Emotional Support Conversation


**Paper Topic And Main Contributions:**

This paper aims to construct a polite interpersonal psychotherapy dialogue system by considering the user's gender, age, persona, and sentiment. The authors used GPT-J as the base model to construct a psychotherapy dialogue dataset in a few-shot manner. The dataset is annotated with the user's profile, i.e., gender, age, and persona, at the dialogue-level, and is annotated with the user's sentiment and therapist’s politeness and interpersonal behavior at the utterance. Based on this dataset, the authors trained a response generation model on GPT-2 using fine-tuning and policy fine-tuning.

**Questions For The Authors:**

(1) On which part of the data are Attribute-specific Reward Models trained? How to split data to get input-label pairs? Is each turn of data used as training data to predict gender, age, and persona?

(2) I am still confused about how to determine the weight of each reward, the authors only mentioned using heuristic approaches in the Limitations. Also, how stable is the training?


**Reasons To Accept:**

This paper contributes a psychotherapy data resource distilled from GPT-J with fine-grained user-side and therapist-side annotations.

**Reasons To Reject:**

(1) This paper lacks the elaboration and analysis statistics of some details. This paper claims the importance of politeness. The examples shown in Figure 1 are all differences in words and sentence patterns. This paper does not count the diversity of polite expressions in the collected data and the differences among different gender, age, and person. Furthermore, the presented examples show that politeness is not specific to gender, age, and persona. In terms of presentation, the term "persona" generally includes "gender" and "age".

(2) The paper misses essential references, e.g., ESConv, EmpatheticDialogues, AugESC, etc. The "polite" proposed in this paper is similar to empathetic phrases, and the "interpersonal behavior" is similar to response strategies, but this paper does not compare the differences and advantages between the constructed dataset and existing datasets (e.g., ESConv, EmpatheticDialogues, etc.). In addition, the data construction method of this paper is similar to AugESC, but the difference and correlation with AugESC in construction methods are not compared.

(3) Ethical considerations. Some of the psychological issues involved in this paper have a higher level of ethical risk. Although the authors have given an ethical statement, limited by the ability of GPT-J, it is still difficult to guarantee the ethical risk issue of the data in a few-shot manner. Why not use a more secure and controllable data distillation model like ChatGPT?

(4) Multiple fine-grained rewards are set in the model proposed in this paper, but there is no relevant experiment to verify how much each reward contributes to the overall performance.

[1] Towards Empathetic Open-domain Conversation Models: a New Benchmark and Dataset
[2] Towards Emotional Support Dialog Systems
[3] AugESC: Dialogue Augmentation with Large Language Models for Emotional Support Conversation


**Reproducibility:**

3: Could reproduce the results with some difficulty. The settings of parameters are underspecified or subjectively determined; the training/evaluation data are not widely available.

**Reviewer Confidence:**

5: Positive that my evaluation is correct. I read the paper very carefully and I am very familiar with related work.

---

> ### Author Rebuttal · Authors · 2023-08-29
>
> 1. This paper lacks the elaboration and analysis statistics of some details. This paper claims the importance of politeness. The examples shown in Figure 1 are all differences in words and sentence patterns. This paper does not count the diversity of polite expressions in the collected data and the differences among different gender, age, and person. Furthermore, the presented examples show that politeness is not specific to gender, age, and persona. In terms of presentation, the term "persona" generally includes "gender" and "age".
>
> Response1: Thank you for highlighting the importance of politeness diversity. We understand that our initial presentation might have fallen short in capturing the nuances of these factors. We have indeed conducted analyses to observe how politeness varies based on different attributes such as gender and age. For example, our findings show that conversations generated for Gender=female and Age=elderly tend to exhibit higher levels of politeness.  In our revised manuscript, we will provide a more thorough analysis of the diversity of polite expressions in our collected data, along with a nuanced exploration of differences across gender, age, and persona. We would also like to clarify here that by term “persona”, we mean here BIG FIVE OCEAN personality traits, of which one is considered to generate conversations. For instance, it has been found that for extraversion persona trait the generated conversations had been much more polite as compared to other traits. We will ensure clarity in our revised manuscript on “persona” aspect analysis as well.
>
> 2. The paper misses essential references, e.g., ESConv, EmpatheticDialogues, AugESC, etc. The "polite" proposed in this paper is similar to empathetic phrases, and the "interpersonal behaviour" is similar to response strategies, but this paper does not compare the differences and advantages between the constructed dataset and existing datasets (e.g., ESConv, EmpatheticDialogues, etc.). In addition, the data construction method of this paper is similar to AugESC, but the difference and correlation with AugESC in construction methods are not compared.
>
> Response2: We apologize for the oversight in missing references to existing works like ESConv, EmpatheticDialogues, and AugESC. In our revised manuscript, we will perform a comprehensive comparative analysis, highlighting the differences, similarities, and benefits of our dataset and approach in relation to these existing datasets. Specifically, our data construction method diverges from AugESC, particularly in terms of the prompt provided to the LLM. We fed <task_description> - consisting of task, psychological issue, and psychotherapeutic approach that had to be followed by Therapist in assisting the client, <seed_utterances>, and user profile aspects: gender, age, persona as prompt to the GPT-J. Lastly, polite and interpersonal behavior therapy annotations were done only after dialogue generation and cleaning steps, differing from AugESC, which used EmpatheticDialogues to extract out the negative emotion reflecting information and posts for dialogue completions.
>
> 3. Ethical considerations. Some of the psychological issues involved in this paper have a higher level of ethical risk. Although the authors have given an ethical statement, limited by the ability of GPT-J, it is still difficult to guarantee the ethical risk issue of the data in a few-shot manner. Why not use a more secure and controllable data distillation model like ChatGPT?
>
> Response3: We appreciate your concern about the ethical risks associated with psychological issues. We understand the potential ethical risks associated with data distillation, even with manual oversight. It is important to note that we opted for GPT-J due to its accessibility and availability for researchers. However, we are aware of the limitations of GPT-J and have taken steps to mitigate ethical risks by involving linguistics and psychology experts for manual validation and corrections. This process is done to ensure high quality psychotherapeutic conversation without any leakage of sensitive information. Lastly, we would provide our dataset only for research purposes after considering a proper request. We will revise our ethical considerations section to provide more clarity on the steps taken to mitigate ethical risks.
>
> 4. Multiple fine-grained rewards are set in the model proposed in this paper, but there is no relevant experiment to verify how much each reward contributes to the overall performance.
>
> Response4:  We acknowledge the need to validate the contributions of individual fine-grained rewards to the overall model performance. To achieve this, we have performed experiments with varying weights (β_1, β_2, β_3, β_4, β_5, γ_1, γ_2, δ_1, δ_2) for each reward. By systematically adjusting these weights and evaluating model performance, we can determine the impact of each reward on the overall outcome. Our results and analyses for these experiments are presented in Table 8 of the appendix to provide transparency and insights into the contribution of each reward.
>
> 5. On which part of the data are Attribute-specific Reward Models trained? How to split data to get input-label pairs? Is each turn of data used as training data to predict gender, age, and persona?
>
> Response5: We performed all experiments with a train:test split of 80:20 and whole data is annotated with required input-label pairs. Hence, attribute specific reward model is trained on this 80% of the data by tracking attribute reward R_A value for each turn, i.e. at each turn, for a generated Therapist’s response, its corresponding reward R_A is computed which is optimized to be maximized. We will provide a more detailed description of this process in our revised manuscript.
>
> 6. I am still confused about how to determine the weight of each reward, the authors only mentioned using heuristic approaches in the Limitations. Also, how stable is the training?
>
> Response6: Each reward weight’s value is kept between 0 to 1 such that all weights sum to 1. This gave us freedom to experiment with different weights for reward models R_A, R_R and R.
> To determine the best combination of β_1, β_2, β_3, β_4, β_5, γ_1, γ_2, δ_1, and δ_2, we adopted a heuristic approach. We start with high values for R_R ( β_1, β_2, β_3, β_4, β_5) reward weights, and increase it until and unless model performance decreases. In subsequent steps on the model's performance decrease, we start giving values to R_A reward (γ_1, γ_2) weights with different combinations like, giving high values to only one single reward and checking the spike in decrease/increase in model’s performance. Then, we check the importance of R_A (δ_1) and R_R (δ_2). Lastly, the best performing combination of weights (β_1, β_2, β_3, β_4, β_5, γ_1, γ_2, δ_1, and δ_2) is selected. We will elaborate on our heuristic approach for setting reward weights and discuss the stability of training in our manuscript. We will ensure that our explanations provide clearer insights into these aspects.
>
> 7. Missing References:
> [1] Towards Empathetic Open-domain Conversation Models: a New Benchmark and Dataset [2] Towards Emotional Support Dialog Systems [3] AugESC: Dialogue Augmentation with Large Language Models for Emotional Support Conversation
>
> Response7: Thank you for bringing up the missing reference. We apologize for any oversight and will promptly include all these references in camera-ready version.

---

### Official Review · Reviewer_jwGD · 2023-08-05

**Soundness:** 3

**Excitement:**

4: Strong: This paper deepens the understanding of some phenomenon or lowers the barriers to an existing research direction.

**Paper Topic And Main Contributions:**


The paper presents a novel model for interactions
in the Psychotherapy settings.
To this end, the authors suggest (based on previous works
and other resources that for effective communication) the incorporation of
user factors such as age, persona (extroversion, neurotic etc), gender
as well as specific
Psychotherapeutic approaches
adopted from (directive, non-directive, eclectic) into the interaction. Moreover, at the utterance-level it is
important to be polite, account for sentiment in previous
utterance and employ a variety of interpersonal
behavioral approaches (such as confrontational, helpful, empathetic etc).

As part of their contributions, they created a novel dataset for
Psychotherapy conversations (using GPT-J) along with expert
inputs and corrections as well as annotations at dialog and utterance
levels. In addition, they propose an RL-based model that uses
rewards and penalties based on the various user and response aspects.
Comparison with competitive baselines shows the effectiveness of their
e-Therapist model for incorporating the
various politeness, Psychotherapeutic styles, sentiment constraints
essential in Psychotherapy.


**Questions For The Authors:**

Also listed in previous section.

C. The numbers in Table-1 are using the Roberta classifiers? (
From 548, it appears to be one of the weighted accuracy or F1, please
clarify). Then considering these classifiers were fine-tuned on smallish
datasets, how reliable are these numbers for comparing across the baselines?

D. Not sure what is n in line 531 and why the selection is what it is.


**Reasons To Accept:**

A. The paper presents an advancement in handling Psychotherapy-related
dialog through its contributions towards creation of a relevant dataset
as well as the incorporation of various conversational elements
essential to Pschotherapy sessions in a dialog generation model via
reinforcement learning.

B. Overall the paper is well-written although there are too many
abbreviations and terminology to make it somewhat hard to track and read. The motivation
and the proposed methods have merit and the experiments are set up well
to showcase their benefits compared to the state-of-the-art baselines for
this topic.


**Reasons To Reject:**

A. The motivating use-case and the name (e-Therapist)
are not very representative. A therapist is expect to advice on
how to address specific problem of the user with relevant
therapy-grounded solutions. But both in the anecdotal examples,
seed utterances, and from the overall reading of the paper it appears
that the proposed model does not really go there (and
only seems to be handling some sort of gathering
preliminary information=problem elicitation state in therapy sessions).

B. Although there is some mention of involvement of a Psychotherpist
as a guide, the dataset creation was primarily by non-experts
in Psychotherapy. Indeed they are Linguistic students, so while some
generic politeness and sentiment annotations are OK,
would it be OK to use Psychotherapeutic labels provided by
these annotators? In the dataset creation, how were the cases selected
from Reddit?
Was any triaging done? These details are completely missing except
for brief mentions in lines 258-259. I also find the anecdotal motivating examples a bit unrealistic and machine-like in Figure-1 ("Hey girl/Hello dear").


**Reproducibility:**

4: Could mostly reproduce the results, but there may be some variation because of sample variance or minor variations in their interpretation of the protocol or method.

**Reviewer Confidence:**

3: Pretty sure, but there's a chance I missed something. Although I have a good feel for this area in general, I did not carefully check the paper's details, e.g., the math, experimental design, or novelty.

---

> ### Author Rebuttal · Authors · 2023-08-29
>
> 1. The motivating use-case and the name (e-Therapist) are not very representative. A therapist is expect to advice on how to address specific problem of the user with relevant therapy-grounded solutions. But both in the anecdotal examples, seed utterances, and from the overall reading of the paper it appears that the proposed model does not really go there (and only seems to be handling some sort of gathering preliminary information=problem elicitation state in therapy sessions).
>
> Response1: We acknowledge the observation regarding our motivating use-case and the name "e-Therapist." We understand the reviewer's perspective on how the use-case might primarily seem to involve preliminary information gathering. However, we would like to emphasize that while the PSYCON dataset does indeed encompass instances where therapists assist users in the initial stages of problem elicitation, it also encompasses diverse conversational contexts where therapists address specific issues of patients and guide them through various therapy sessions. For instance, we would like to highlight Conversation 1 from our sample dataset, available at the link: https://anonymous.4open.science/r/Anon-2AAD/. In this conversation, a therapist guides a patient through deep breathing exercises, offers relaxation techniques, and addresses the patient's sleep and nutrition patterns. This example illustrates that our e-Therapist model does encompass specific issue-based interactions as well. Example:
>
> Patient:	No, I haven't tried anything like that.
>
> Therapist:	Okay, well let's start by practising some deep breathing right now. Take a deep breath in through your nose, hold it for a few seconds, and then slowly exhale through your mouth. Let's do that a few more times together.
>
> Patient:	deep breath in and exhale That does feel a bit better.
>
> Therapist:	Good, I'm glad to hear that. You can use this technique anytime you're feeling stressed or overwhelmed. And I can also recommend some other relaxation exercises that might be helpful for you.
>
> Patient” 	That would be great, thank you.
>
> Therapist:	Now, let's also talk about some ways to improve your overall mental health. Have you been getting enough sleep and eating well?
>
> Patient:	Not really. I've been having trouble sleeping and I haven't been eating very healthily.
>
> Therapist:	Okay, those are two really important things to work on. Lack of sleep and poor nutrition can both contribute to feelings of depression and anxiety. Let's start by setting a goal to get at least 7-8 hours of sleep each night. And when it comes to nutrition, try to focus on incorporating more fruits, vegetables, and whole grains into your diet.
>
> Patient:	That sounds like a good plan.
>
> Therapist:	Great. And lastly, I want to touch on the importance of being kind to yourself. Remember that recovery from depression takes time and it's okay to have setbacks along the way. What's most important is that you keep working towards your goals and continue to take care of yourself.
>
> 2. Although there is some mention of involvement of a Psychotherpist as a guide, the dataset creation was primarily by non-experts in Psychotherapy. Indeed they are Linguistic students, so while some generic politeness and sentiment annotations are OK, would it be OK to use Psychotherapeutic labels provided by these annotators? In the dataset creation, how were the cases selected from Reddit? Was any triaging done? These details are completely missing except for brief mentions in lines 258-259. I also find the anecdotal motivating examples a bit unrealistic and machine-like in Figure-1 ("Hey girl/Hello dear").
>
> Response2: We understand the importance of expertise in psychotherapy when creating a dataset for training our model. While linguistic students were indeed involved in annotation, we actively collaborated with experienced psychotherapists to ensure the appropriateness and quality of the labels.  We have not done any triaging in terms of urgency, severity, or other likewise parameters, but we have extracted seed utterances from Reddit using keywords like depression, anxiety, stress, bipolar disorder, disruptive behaviour and dissocial disorders, post-traumatic stress disorder etc.. After extracting these seed utterances these are manually checked in terms of appropriateness ( of which 35% were dropped as they were not good enough to be treated as seed utterances and approx. 40% were linguistically corrected to align the flow of language as if a therapeutic conversation is starting). Additionally, we have taken the important step of incorporating three psychotherapy approaches (directive, non-directive, and eclectic) within the prompt generation, leading to diverse therapy conversations for each set of seed utterances. We will clarify this in camera-ready version by providing a more comprehensive explanation in the relevant section.
>
> 3. The numbers in Table-1 are using the Roberta classifiers? ( From 548, it appears to be one of the weighted accuracy or F1, please clarify). Then considering these classifiers were fine-tuned on smallish datasets, how reliable are these numbers for comparing across the baselines?
>
> Response3: The W-ACC and F1 scores of GAC, PC, SC, CTC, PoC, and IBC classifiers are computed on 80:20, train:test split of PSYCON dataset with manually annotated 340 dialogues which are shown in Table 7. In Table 1, we computed all the metrics based on W-ACC on a whole dataset of 1020 conversations with all annotated labels.
>
> 4. Not sure what is n in line 531 and why the selection is what it is.
>
> Response4: In this context, 'n' represents the number of candidate responses generated by the SLLM and subsequently rewarded using the reward R. To clarify further, these 'n' candidate responses encompass distinct variations of outputs for a given input. These candidates are then ranked based on the reward R. By optimizing the SLLM to generate responses that are highly ranked according to this reward, we create a mechanism that encourages the model to generate relevant outputs. This approach aids in ensuring that the model's generated responses align well with the desired outcomes as ensured by different rewards.

---

### Official Review · Reviewer_YcLm · 2023-08-05

**Soundness:** 4

**Excitement:**

3: Ambivalent: It has merits (e.g., it reports state-of-the-art results, the idea is nice), but there are key weaknesses (e.g., it describes incremental work), and it can significantly benefit from another round of revision. However, I won't object to accepting it if my co-reviewers champion it.

**Paper Topic And Main Contributions:**

This paper addresses the problem of dialogue systems for interactions between users and therapists by including the personalized information of the user, like gender, age, persona, and sentiment. For this purpose, they create a novel conversational dataset, “PSYCON,” that contains annotations in dialogue level like gender, age, persona, and psychotherapeutic approach. Furthermore, the dataset also contains utterance-level annotations like sentiment, politeness, and interpersonal behavior information.  Then, the paper introduces e-Therapist, an interpersonal dialogue system fine-tuned in a RL environment. The results show the improvement in dialogue experience by incorporating interpersonal attributes.

**Reasons To Accept:**

- The dataset collected is a valuable resource for future work, and the authors combine LLM prompting with expert interventions to lower the costs and control the quality of the dataset.
- A novel dialogue system that utilizes a RL framework for fine-tuning.

**Reasons To Reject:**

- It is not clear how significant the results are. In addition, the effect of different components of the reward functions is not explored. While the authors explore by setting to 0 each R, RA, or RR, they do not address the effects of their components like R1-7.

**Reproducibility:**

4: Could mostly reproduce the results, but there may be some variation because of sample variance or minor variations in their interpretation of the protocol or method.

**Reviewer Confidence:**

4: Quite sure. I tried to check the important points carefully. It's unlikely, though conceivable, that I missed something that should affect my ratings.

**Typos Grammar Style And Presentation Improvements:**

Suggestion: I would probably recommend comparing your model against a bigger language model (like GPT-J), where you utilize a few shot-learning.

---

> ### Author Rebuttal · Authors · 2023-08-29
>
> 1. It is not clear how significant the results are. In addition, the effect of different components of the reward functions is not explored. While the authors explore by setting to 0 each R, RA, or RR, they do not address the effects of their components like R1-7.
>
> Response1: We appreciate your observation regarding the significance of our results and the exploration of different rewards. While we agree that solely setting R, R_A, and R_R to 0 might not provide a complete understanding, we took a comprehensive approach to address this concern. Our experiment involved systematically varying weight values (β_1, β_2, β_3, β_4, β_5, γ_1, γ_2, δ_1, δ_2) assigned to each rewards. By doing so, we quantified the influence of each of the rewards and their combinations more comprehensively. The outcomes of these experiments, presented in Table 8 within the appendix, offer a holistic view of our findings. This approach allows us to draw more robust conclusions about the effects of each of the rewards on the overall model performance.
>
> 2. Suggestion: I would probably recommend comparing your model against a bigger language model (like GPT-J), where you utilize a little shot-learning.
>
> Response2: We appreciate the valuable suggestion to compare our model against a larger language model, such as GPT-J. While we acknowledge the significance of this approach, we regret to inform you that our computational resources were, unfortunately, limited in this regard. It's possible to do inference using GPT-J for dataset creation, but our ability to fine-tune with a GPT-J-like model was hampered by the constraints of a 40GB GPU. It became apparent that the scope of our input conversations, averaging 27 turns (even some exceeds to 45 turns) with around 30 tokens per response, resulted in a cumulative token count of approximately 810 tokens (maximum 1350 tokens). Regrettably, this far exceeded the capacity of our available resources. These constraints highlight the practical challenges we encountered during our research. Despite these limitations, we assure you that we have strived to maximize the potential of our available resources to produce meaningful results. On positive side, we have requested for resources and would try our best to show the experimental results with GPT-J in camera-ready version of the paper.

---

### Meta-Review · Area_Chair_ggQW · 2023-09-18

**Recommendation:** 4

**Metareview:**

This work constructs a polite interpersonal psychotherapy dialogue system by conditioning system responses on the user's gender, age, persona, and sentiment.

Contributions:
 - The (synthetic) dataset collected is a valuable resource for future work.

Weakness:
 - As presented it is not clear how significant the results are, there is a lack of analysis of some details, and in relation to the main claim, which is the importance of politeness the examples do not seem to show a specific connection to gender, age, and persona.

---

### Decision · Program_Chairs · 2023-10-07

**Decision:**

Accept-Main

**Comment:**

This work constructs a polite interpersonal psychotherapy dialogue system by conditioning system responses on the user's gender, age, persona, and sentiment.

Contributions:
 - The (synthetic) dataset collected is a valuable resource for future work.

Weakness:
 - As presented it is not clear how significant the results are, there is a lack of analysis of some details, and in relation to the main claim, which is the importance of politeness the examples do not seem to show a specific connection to gender, age, and persona.